# English–Welsh Cross-Lingual Embeddings

Luis Espinosa-Anke [1,*,†], Geraint Palmer [2,†], Padraig Corcoran [1], Maxim Filimonov [1], Irena Spasić [1] and Dawn Knight [3]

1   School of Computer Science and Informatics, Cardiff University, Cardiff CF24 3AA, UK;
    corcoranp@cardiff.ac.uk (P.C.); filimonovm@cardiff.ac.uk (M.F.); spasici@cardiff.ac.uk (I.S.)
2   School of Mathematics, Cardiff University, Cardiff CF24 4AG, UK; palmergi1@cardiff.ac.uk
3   School of English, Communication and Philosophy, Cardiff University, Cardiff CF10 3EU, UK;
    knightd5@cardiff.ac.uk
*   Correspondence: espinosa-ankel@cardiff.ac.uk
†   These authors contributed equally to this work.

**Abstract:** Cross-lingual embeddings are vector space representations where word translations tend to be co-located. These representations enable learning transfer across languages, thus bridging the gap between data-rich languages such as English and others. In this paper, we present and evaluate a suite of cross-lingual embeddings for the English–Welsh language pair. To train the bilingual embeddings, a Welsh corpus of approximately 145 M words was combined with an English Wikipedia corpus. We used a bilingual dictionary to frame the problem of learning bilingual mappings as a supervised machine learning task, where a word vector space is first learned independently on a monolingual corpus, after which a linear alignment strategy is applied to map the monolingual embeddings to a common bilingual vector space. Two approaches were used to learn monolingual embeddings, including word2vec and fastText. Three cross-language alignment strategies were explored, including cosine similarity, inverted softmax and cross-domain similarity local scaling (CSLS). We evaluated different combinations of these approaches using two tasks, bilingual dictionary induction, and cross-lingual sentiment analysis. The best results were achieved using monolingual fastText embeddings and the CSLS metric. We also demonstrated that by including a few automatically translated training documents, the performance of a cross-lingual text classifier for Welsh can increase by approximately 20 percent points.

**Keywords:** natural language processing; distributional semantics; machine learning; language model; word embeddings; machine translation; sentiment analysis

## 1. Introduction

A popular research direction in current natural language processing (NLP) research consists of learning vector space representations of words for two or more languages, and then applying some kind of transformation to one of the spaces such that "cross-lingual synonyms", i.e., words with the same meaning across languages, are assigned similar vector space representations. The applications of these cross-lingual embeddings into downstream tasks is indisputable today, ranging from information retrieval [1], entity linking [2], text classification [3,4], as well as natural language inference or lexical semantics [5]. These cross-lingual embeddings are often learned and evaluated for language pairs, for which there is either a good availability of parallel or comparable text corpora, supervision signal, or, at the least, large enough raw but non-aligned corpora for each language (see, e.g., Mikolov et al. [6], Conneau et al. [7], Artetxe et al. [8,9]).

However, the availability of cross-lingual mappings between resource-rich and resource-poor languages still constitutes a challenge [10]. In this paper, we are particularly concerned with learning cross-lingual embeddings between the languages of English and Welsh. The last census indicated that there are currently 526,016 speakers of Welsh (https://gov.

wales/welsh-language-data-annual-population-survey-2019, accessed on 15 July 2021).
Welsh is statistically a 'minority' language as there are more speakers of English than Welsh
in Wales and the UK, but it is a healthy one, and Wales represents the largest bilingual com-
munity in the UK. As a minoritized language in the UK context (albeit with official status,
alongside English, in the devolved nation of Wales), Welsh does not enjoy the same lan-
guage technology resources as English or other major state languages, although there is an
increasing interest in widening the availability of resources in this context. Welsh-language
technologies that are currently available include POS (part of speech) taggers (including Cy-
Tag, [11]), WordNet Cymru (https://users.cs.cf.ac.uk/I.Spasic/wncy/index.html, accessed
on 15 July 2021), and an extensive range of tools developed for the purposes of, for exam-
ple, text-to-speech, speech recognition, machine translation, and terminology recognition,
developed by Canolfan Bedwyr at Bangor University (see their online Welsh National Lan-
guage Technologies Portal (http://techiaith.cymru/?lang=en, accessed on 15 July 2021).
However, to the best of our knowledge, there has been no work on learning high-quality
bilingual mappings between English and Welsh, which would drastically accelerate the
current landscape for Welsh NLP technologies. In this paper, we thus propose to explore
current state-of-the-art cross-lingual embeddings techniques for the Welsh language. We
first train several monolingual models based on Skip-gram [12] and fastText [13], consider-
ing several configurations in terms of context window size, minimum frequency threshold,
and vector dimensionality. Then, we apply VecMap [14], a method for learning cross-
lingual mappings via orthogonality constraints to our monolingual embeddings. We also
report results on a post-processing step based on applying an additional transformation
obtained via a linear model trained on top of the bilingual synonym's mean vectors [5].
These cross-lingual representations are evaluated on the standard task of dictionary induc-
tion. Finally, as a further downstream task, we report the results of a sentiment analysis
system for Welsh in zero-shot and few-shot settings, i.e., training it only with English data,
or with limited instances of (automatically translated) task-specific Welsh data. Our results,
while promising, also point to the challenges posed by under-represented, resource-poor
languages in NLP development, and suggest that further research is needed to strengthen
the landscape for Welsh language technologies. The contributions of this paper are as
follows:

- Cross-lingual embeddings: we train, evaluate, and release a wealth of cross-lingual English–Welsh word embeddings.
- Train and test dictionary data: we release to the community a bilingual English–Welsh dictionary with a fixed train/test split, to foster reproducible research in Welsh NLP development.
- Sentiment analysis: we train, evaluate, and release a Welsh sentiment analysis system, fine-tuned on the domain of movie reviews.
- Qualitative analysis: we analyze some of the properties (in terms of nearest neighbors) of the cross-lingual spaces, and discuss them in the context of avenues for future work.

Our results suggest that gains in training English–Welsh bilingual embeddings can be
obtained by carefully tuning the hyperparameters of the monolingual models, and that
the distance metric chosen matters, with differences of up to 5% in accuracy. Overall,
the best configuration across the board seems to always involve the fastText model (as
opposed to skip-gram), and the CSLS distance metric (as opposed to cosine similarity
and inverted softmax). Conversely, the cutoff threshold for minimum frequency and the
context window seem to be less important for the final results, as there is not a clear pattern
involving a consistent setting among the top-ranked results. In our external evaluation
experiment, namely, zero and few-shot sentiment analysis, we verified that it is indeed
possible to develop a competitive sentiment analysis system for Welsh only using cross-
lingual embeddings and English training data, and that by adding synthetic Welsh training
data (e.g., from a machine-translation engine), the performance of the model increases
as well.

The remainder of this paper is organized as follows: Section 2 gives an account of

research works in different areas relevant to the scope of this paper; Section 3 introduces resources we used for generating cross-lingual embeddings. Section 4 introduces the algorithm used for mapping monolingual embeddings into a shared space. Section 5 presents the results in two (intrinsic and extrinsic) experimental settings. Finally, Section 6 summarizes the main contributions of this work, and outline potential avenues for future work. Data, models, and software are publicly available at (https://github.com/cardiffnlp/en-cy-bilingual-embeddings, accessed on 15 July 2021).

## 2. Background

In this section, we present a review of related works with respect to NLP for the Welsh language and cross-lingual word embeddings.

### 2.1. Welsh Language NLP

Recently there has been much research in the space of applying NLP to non-English minority languages such as Welsh. The defining characteristic of a minority language is that the amount of corresponding data available for that language is significantly less than that available for the English language. Most state-of-the-art NLP models use deep learning where performance scales with the amount of available data. Given this, achieving performance on NLP tasks for minority languages is on par with that achieved for the same tasks for the English language represents a significant challenge. The Welsh Natural Language Toolkit (WNLT) is a Welsh-Government-funded project which focuses on the development of NLP tools for the Welsh language (https://hypermedia.research.southwales.ac.uk/kos/wnlt/, accessed on 15 July 2021). The tools in question are distributed under GNU Lesser General Public License (LGPL) and include tools for tokenization, lemmatization, POS tagging, and Named Entity Recognition (NER) (https://sourceforge.net/projects/wnlt-project/, accessed on 15 July 2021). Neale et al. [11] developed a rule-based part-of-speech (POS) tagger for the Welsh language entitled Cy-Tag. Although state-of-the-art POS taggers for the English language use deep learning, the authors argue there is insufficient Welsh language data to use such an approach for the Welsh language. The same authors later developed a rule-based semantic tagger, entitled CySemTagger [15]. Both of these tools are available under a free software (GPL version 3) licence (https://github.com/CorCenCC, accessed on 15 July 2021). Jones et al. [16] developed a statistical machine translation model for the English and Welsh language pair. Spasić et al. [17] developed a statistical method for multiple word term recognition in Welsh. This method builds on a previously proposed term-recognition method known as FlexiTerm [18].

### 2.2. Cross-Lingual Embeddings

Earlier attempts to train cross-lingual word embeddings required access to parallel (or, at least, comparable) corpora [19–25]. Finding such corpus especially for a minoritized language can prove challenging. Therefore, the research in this space gravitated towards using bilingual dictionaries instead of aligning the words in respective languages [6,26]. It was later shown that such cross-lingual supervision is not necessary to align word embedding [7]. Instead, adversarial training can be used to initialize a linear mapping between two vector spaces and produce a synthetic parallel dictionary. The success of this approach was based on the use of two metrices: one for unsupervised validation and the other one for similarity measure. Such combination reduces the hubness problem while improving the translation accuracy.

Hubness is a phenomenon that occurs in high-dimensional spaces, where some objects tend to concentrate around a centroid while others have few nearest neighbors [27]. Specifically, hubness associated with cross-lingual embeddings was explored in [28], who proposed incorporating a nearest-neighbor reciprocity as a way of managing hubness. Different measures were used to down-weigh similarities associated with hub words, including cross-domain similarity local scaling (CSLS) [7] and inverted softmax [29]. In ad-



dition, adding an orthogonality constraint, which conveniently has a closed-form solution, can improve performance further [30].

Alternatively, to align monolingual embedding spaces with no supervision, Zhang et al. [31] used adversarial training to exploit sudden drops in accuracy for model selection followed by minimizing the earth-mover distance [32]. Conversely, Conneau et al. [7] do not base model selection on its performance, which allows for hyper-parameters to be tuned specifically for a given language pair as they tend to vary significantly across languages. Similar approaches used to induce bilingual dictionaries from data [5,10,14,33] yielded state-of-the-art performance in many language pairs, although the experimental setup followed in the literature has also been closely scrutinized, and there exist studies that argue for experiments that account for different genres in source and target corpora, studying (dis)similarities between languages, etc. [34,35].

Although the advent of language models in the current NLP landscape (BERT, GPT, or RoBERTa) [36–38] has transformed the field, it is also true that even for languages where the availability of raw data is small, having access to pre-trained static word embeddings can make the difference between developing a language technology or not at all. Recent work has, for example, focused on dialectal Arabic, by combining BERT-based encodings with Arabic word embeddings for underrepresented domains and dialects [39].

## 3. Materials

This section describes the materials required for generating cross-lingual embeddings.

### 3.1. Corpora

While a number of Welsh corpora exist, there generally lacks extensive data sets of Welsh language that are freely/widely available. To undertake this study, we combined a number of existing Welsh corpora, sourced from different language contexts, including proceedings from the Welsh assembly (http://cymraeg.org.uk/kynulliad3/, accessed on 15 July 2021), scraped websites and blogs [40] and the National Corpus for Contemporary Welsh (CorCenCC, [41]), amongst others. The full list of corpora used are given in Table 1. We ensured that the collected corpus includes a diverse range of formats, genres and registers, including a balanced mix of formal and casual language, and general and specialized topics. For example, there are texts from the highly formal academic writing of academic journal papers and textbooks; the archaic writing of the bible; technical writing in the form of administrative documents and software documentation; journalistic writing from news and magazine articles; pieces of creative writing in prose, poetry and song; and everyday casual language including emails, tweets, text messages, and transcripts of spoken language.

In terms of the English corpus, we used a Wikipedia data dump for June 2018, which is a standard corpus in distributional semantics for learning word embeddings.

### 3.2. Text Corpus Creation

We developed Welsh and English corpora to train our bilingual embeddings, drawing on a range of pre-existing data sets. The full Welsh-language data set extended to 144,976,542 words after tokenization. The names and corresponding number of words in each individual text corpus are displayed in Table 1. We now provide a brief description of each individual text corpus.

**Table 1.** Names and corresponding number of words in each individual Welsh-language text corpus.

| Corpus | Numb. Words |
| --- | --- |
| Welsh Wikipedia | 21,233,177 |
| Proceedings of the Welsh Assembly 1999–2006 | 11,527,963 |
| Proceedings of the Welsh Assembly 2007–2011 | 8,883,870 |
| The Bible | 749,573 |
| OPUS translated texts | 1,224,956 |
| Welsh Government translation memories | 1,857,267 |
| Proceedings of the Welsh Assembly 2016–2020 | 17,117,715 |
| Cronfa Electroneg o Gymraeg | 1,046,800 |
| An Crúbadán | 22,572,066 |
| DECHE | 2,126,153 |
| BBC Cymru Fyw | 14,791,835 |
| Gwerddon | 732,175 |
| Welsh-medium websites | 7,388,917 |
| CorCenCC | 10,630,657 |
| S4C subtitles | 26,931,013 |

**Welsh Wikipedia**—Wikipedia is a multilingual crowd-sourced online encyclopedia and one of the world's most popular websites. English Wikipedia was the first edition of Wikipedia and was founded in January 2001. As of 29 September 2019 (when these data were collected), there were 5,938,555 articles contained in this project. Given the large number of articles, English Wikipedia is a text corpus commonly used to train English language word embeddings. Welsh Wikipedia is the Welsh language edition of Wikipedia and was founded in July 2003. It is significantly smaller than English Wikipedia and as of 29 September 2019 it contains 106,128 articles. Web crawling of this was undertaken, specifically, using the Python library urllib and the Python library Beautiful Soup to extract all text within paragraph tags <p>. We subsequently removed all citations and mathematical equations.

**National Assembly for Wales 1999–2006**—The National Assembly for Wales is the devolved parliament of Wales, which has many powers, including those to make legislation and set taxes. By performing a web crawling of the Assembly website (http://xixona.dlsi.ua.es/corpora/UAGT-PNAW/, accessed on 15 July 2021), Jones et al. [16] created a bilingual aligned corpus of Welsh and English from the online version of the Proceedings of the Plenary Meetings of the Assembly between the years 1999 and 2006 inclusive. This is freely available as a plain text file. Only the Welsh part of this corpus was used for the purposed of the current project.

**National Assembly for Wales 2007–2011**—Donnelly [42] created the Kynulliad3 corpus, which is similar to the previous bilingual aligned corpus except that it covers the period between the years 2007 and 2011 inclusive. This corpus, which contains 350,000 aligned Welsh and English sentences, was extracted by querying an SQL database. Only the Welsh half of this corpus was used in the current project.

**The Bible**—Beibl.net (http://www.beibl.net, accessed on 15 July 2021) includes all books of the Bible in modern Welsh. Texts were scraped using urllib and Beautiful Soup in Python.

**OPUS**—OPUS is a collection of technical texts on the web, mainly including software documentation, in a number of languages. We extracted a range of en-cy (English–Welsh) texts from this resource in plain text format.

**Welsh Government translations memories**—The collection of translation memory files contains published bilingual documents and other materials from the Welsh Government (from August 2019 to May 2020). The data set comprises .tmx files, which were extracted using Python's translate toolkit package.

**National Assembly for Wales 2016–2020**—Records of the proceedings of the Welsh Assembly, including plenary information from the start of the Fifth Assembly (May 2016) and

Committee information from November 2017 to May 2020. The data set was downloaded as .xml, with text extracted using the Python library Beautiful Soup.

**Cronfa Electroneg o Gymraeg**—This corpus contains 500 articles of approximately 2000 words each, selected from a representative range of text types to illustrate modern (mainly post-1970) Welsh prose writing [43]. It includes articles from the fields of novels and short stories, religious writing, children's literature, non-fiction materials in the fields of education, science, business and leisure activities, public lectures, newspapers and magazines, reminiscences, academic writing, and general administrative materials (letters, reports, minutes of meetings).

**An Crúbadán**—This corpus was created by Scannell [40] by performing web crawling. It consists of a collection of Welsh Wikipedia articles, Welsh Tweets, Welsh Blogs, the Universal Declaration of Human Rights, and articles from a Jehovah's Witnesses website (JW.org) (https://www.jw.org/cy/, accessed on 15 July 2021). To prevent duplication of the previous Welsh Wikipedia corpus, we removed all Wikipedia articles.

**DECHE**—The Digitization, E-publishing, and Electronic Corpus (DECHE) project produces e-books out of Welsh language scholarly, academic books which are out of print and unlikely to be reprinted in traditional paper format [44]. Candidates for producing as e-books are nominated by lecturers working through the medium of Welsh and prioritized by the Coleg Cymraeg Cenedlaethol, who fund the project. We constructed a corpus from this project by manually downloading all books in epub format and extracting the plain text using the Python libraries epub_conversion and Beautiful Soup.

**BBC Cymru Fyw**—BBC Cymru Fyw is an online Welsh language service provided by BBC Wales containing news and magazine-style articles. Using the Corpus Crawler tool (https://github.com/google/corpuscrawler, accessed on 15 July 2021), we constructed a corpus containing all articles published on BBC Cymru Fyw between 1 January 2011 and 17 October 2019 inclusive.

**Gwerddon**—Gwerddon is a Welsh-medium academic e-journal which publishes research in the Arts, the Humanities, and the Sciences (http://www.gwerddon.cymru/, accessed on 15 July 2021). This corpus contains all text in articles contained in 29 editions of this journal. It was constructed by manually downloading the articles in question and extracting the corresponding text using the R programming language package pdftools. Some manual post-formatting was carried out to correct footnotes, etc.

**Welsh-medium websites**—Golwg360 (https://golwg360.cymru, accessed on 15 July 2021) and O'r Pedwar Gwynt (https://pedwargwynt.cymru, accessed on 15 July 2021) are Welsh-medium news websites. PoblCaerdydd (https://poblcaerdydd.com/, accessed on 15 July 2021) and Cylchgrawn Barn (https://barn.cymru/, accessed on 15 July 2021) are Welsh-medium online magazines. This corpus contains all text extracted from articles on these four websites. It was construed by performing web crawling using wget and extracting all relevant text using the Python library Beautiful Soup.

**CorCenCC**—CorCenCC (https://www.corcencc.org, accessed on 15 July 2021) [41] is the National Corpus of Contemporary Welsh (Corpws Cenedlaethol Cymraeg Cyfoes). This corpus contains over 11 million words of spoken, written, and electronic language data sampled from a range of genres, styles, registers, and dialect regions. The pre-processed version of the corpus was made available for use in this project.

**S4C subtitles**—Subtitles kindly received privately (i.e., not publicly available) from the Welsh-language TV channel S4C (https://www.s4c.cymru, accessed on 15 July 2021). Text manipulation was used to strip away the formatting and compile this corpus.

English corpora include the UMBC (https://ebiquity.umbc.edu/blogger/2013/05/01/umbc-webbase-corpus-of-3b-english-words/, accessed on 15 July 2021) web-based corpus and Wikipedia (www.wikipedia.org, accessed on 15 July 2021) corpus. UMBC contains over 3 billion words, including blog posts, news stories etc., that have been stripped from the web, cleaned, tokenized and pre-processed. The Wikipedia corpus includes all texts from the English Wikipedia site, with one sentence per line, tokenized, lemmatized, chunked, lower-cased and POS-tagged.

### 3.3. Word Embeddings

In our experiments, we compare two different word embeddings methods, namely, Skip–Gram with Negative Sampling (which we denote as *word2vec*) [12], and fastText [13], which is an improved *word2vec* architecture that accounts for subword information in order to capture morphological and subword information. For each of these two models, we experiment with different hyperparameters, namely, *vector size* (**DIM**), a word's *minimum frequency* threshold (**MF**), and *context window* (**CW**).

### 3.4. Bilingual Dictionary

Our initial bilingual dictionary was provided by Bangor University [45]. It contains over 100,000 bilingual entries, including named entities (e.g., "Alfred the Great"), multi-word terms (e.g., "acquired immunity"), or domain-specific terminology (e.g., for the chemical domain, "2,4-diisocyanato-1-methylbenzene"). For our purposes, we preprocessed this initial dictionary by removing all multi-word and ambiguous (i.e., words for which there was more than one entry—or *sense*— recorded in the dictionary) terms, and split it into training and test. The final size of this dictionary, which we used for mapping English to Welsh embeddings, and for evaluating these mappings, consisted of 9067 training pairs and 2268 test pairs.

## 4. Methods

Having a bilingual dictionary available makes it viable to cast the problem of learning bilingual mappings as a supervised machine learning task, where given two monolingual corpora, a word vector space is first learned independently for each language. This can be achieved with standard word embedding models such as Word2vec [6], GloVe [46], or fastText [13]. Second, a linear alignment strategy is used to map the monolingual embeddings to a common bilingual vector space. It is worth mentioning that we do not require parallel or comparable corpora to build these multilingual models [47,48], although it has also been shown that the higher the overlap in terms of domain, topic, genre, or linguistic typology, the better the alignments [35,49].

The learning model for these mappings is often a simple linear transformation trained on a bilingual dictionary. In the original paper by Mikolov et al. [6], a matrix $\mathbf{W}$ is trained, which minimizes the following objective:

$$\sum_{i=1}^{n} \|\mathbf{x_i}\mathbf{W} - \mathbf{z_i}\|^2 \tag{1}$$

with $\mathbf{x_i}$ and $\mathbf{z_i}$ being the vector representations of cross-lingual synonyms (i.e., translations) of two words $w_i$ and $z_i$, in two different languages, respectively. After training, the translation $z'$ of any source word $x'$ in the source language can be defined as $z' = \mathrm{argmax}_{z'} d(\mathbf{W}\mathbf{x}, \mathbf{z}')$, with $d(\cdot)$ being a vector distance metric. In this paper, we consider as options for $d(\cdot)$ the following: (1) the well-known cosine similarity (NN); (2) inverted softmax (*invsoftmax*) [29]; and (3) cross-domain similarity local scaling (CSLS) [7]. This task, i.e., the retrieval of cross-lingual synonyms (or word translations) is known as *dictionary induction*, and is considered a good intrinsic testbed for assessing the quality of cross-lingual mappings. In this paper, we report experiments on the test split of the dictionary described in Section 3.4.

## 5. Results

We report results on the test set of our English–Welsh bilingual dictionary. We report these results in terms of *accuracy* (**ACC.**), i.e., we record a true positive only if the nearest neighbor in the mapped space is a translation of the source word. This is a strict measure (as we could have considered, for instance, $P@k; k \in \{1, 5, 10\}$), which serves as a strong baseline for upcoming research in English–Welsh crosslingual language technologies.

## 5.1. Quantitative Evaluation

The task of bilingual dictionary induction, a natural byproduct of learning bilingual mappings, and which we have introduced in Section 4, is a good proxy for evaluating the quality of cross-lingual mappings.

We thus report results of appplying the VecMap method. However, we also experimented with Meemi, but since the results were slightly lower across most configurations, we only report VecMap performance. Table 2 shows the top 20 configurations in terms of accuracy. As we can see, *fastText* consistently performs best when compared to *word2vec*, and CSLS clearly outperforms inverted softmax and cosine similarity in terms of retrieval metrics. On the other hand, the threshold for minimum frequency and context windows seem to be less relevant, as there is high variability among the best configurations. Regarding the overall scores, note that these are in line with what previous work has found when dealing with language pairs involving English and a low-resource language. For example, Doval et al. [49] report P@1 scores for their best models of 24.8 for English–Finnish, 21.5 for English–Farsi, or 19.3 for English–Russian, and Xu et al. [50] report roughly similar or worse results for dictionary induction experiments involving, e.g., Turkish (9.96) or Latvian (13.53). Note that theirs is an unsupervised approach.

**Table 2.** Top 20 configurations (ranked in descending order) in terms of accuracy (**Acc.**) for the bilingual dictionary induction task *when using VecMap*. We compare different monolingual embedding models (**Model**), vector size (**Dim.**), minimum frequency threshold (**MF**), context window (**CW**), and neighbor retrieval method (**Retrieval**, cf. Section 3).

| Model | Dim. | MF | CW | Retrieval | Acc. |
|:---:|:---:|:---:|:---:|:---:|:---:|
| fastText | 500 | 6 | 6 | CSLS | 22.92 |
| fastText | 500 | 6 | 4 | CSLS | 21.85 |
| fastText | 500 | 6 | 8 | CSLS | 21.75 |
| word2vec | 300 | 6 | 4 | CSLS | 21.75 |
| word2vec | 500 | 6 | 8 | CSLS | 21.46 |
| word2vec | 300 | 6 | 6 | CSLS | 21.46 |
| word2vec | 500 | 6 | 4 | CSLS | 21.46 |
| word2vec | 300 | 6 | 8 | CSLS | 21.36 |
| word2vec | 500 | 6 | 6 | CSLS | 21.36 |
| fastText | 500 | 3 | 4 | CSLS | 20.46 |
| fastText | 500 | 3 | 8 | CSLS | 19.75 |
| fastText | 500 | 3 | 6 | CSLS | 19.36 |
| word2vec | 300 | 3 | 8 | CSLS | 19.22 |
| word2vec | 500 | 3 | 6 | CSLS | 19.22 |
| word2vec | 500 | 3 | 8 | CSLS | 19.18 |
| word2vec | 300 | 3 | 6 | CSLS | 18.83 |
| fastText | 300 | 6 | 4 | CSLS | 18.57 |
| word2vec | 300 | 6 | 4 | invsoftmax | 18.48 |
| word2vec | 300 | 6 | 8 | invsoftmax | 18.48 |
| word2vec | 300 | 6 | 8 | NN | 18.43 |

*5.2. Qualitative Evaluation*

The cross-lingual vector space can be manually explored in order to evaluate how well both the monolingual embeddings capture semantic relationships within a language, and also how well the cross-lingual embeddings align. We start this by selecting a small set of prototype words in the first language, and inspect their nearest neighbors in the second language. We then compare this to the reverse procedure: selecting the same translated, words in the second language, and inspect their nearest neighbors in the first.

Table 3 lists a selection of ten words, and their translations, with their 10 nearest neighbors in their opposite languages. In general, the cross-lingual embeddings align well, with the common nouns, adjectives, and verbs mapping to very similar and very related words in both directions. We also attempted to find closely related words to *hiraeth*, a word often claimed to be untranslatable into English, which still gave accurate nearest neighbors, referring to feelings of longing and yearning for home.

More specialized vocabulary, such as foreign loadwords (*croissant*), and proper nouns (*French*, and place names such as *Cardiff* and *Tonypandy*) show some asymmetry in the alignment of the embeddings. Here, the Welsh nearest neighbors to English words are much more relevant and semantically related than the English nearest neighbors of Welsh words. For example, the Welsh nearest neighbors to *croissant* gives breakfast foods and pastries, while the English nearest neighbors are generic foodstuffs. Similarly, the Welsh nearest neighbors to *French* gives Euro-centric languages and adjectives, while the English nearest neighbors to *ffrangeg* (the French language) gives languages from further afield. It is also interesting to note that in Welsh, the words *ffrangeg* (the French language) is different to *ffrengig* (the French nationality), and all the English nearest neighbors to *ffrangeg* are languages or language-related terms, rather than words related to nationalities, while a mix of the two is seen in the Welsh nearest neighbors of *French*.

Geographic place names are also interesting, with the Welsh nearest neighbors of English place names giving more local and geographically closer place names than the English nearest neighbors of Welsh place names. This may be an effect of the English training corpus having a much more international and broader scope than the Welsh training corpus. For example, *Cardiff* / *caerdydd*, the capital of Wales and thus an important word in the Welsh language: its Welsh nearest neighbors are other major Welsh towns and cities, while it's English nearest neighbors are populated with Australian places, maybe referencing the much smaller Australian town of Cardiff.

**Table 3.** Table of a selected sample of cross-lingual nearest neighbors examples.

| *word_cy* | Closest English Words to *word_cy* | *word_en* | Closest Welsh Words to *word_en* |
|---|---|---|---|
| nofio (*swim*) | swim, swimming, kayak, paddling, rowing, waterski, swam, watersport, iceskating, canoe | swim | nofio (*swim*), deifio (*diving*), cerdded (*walking*), blymio (*diving*), padlo (*paddling*), arnofio (*floating*), sblasio (*splashing*), troelli (*spinning*), neidio (*jumping*) |
| glaw (*rain*) | rain, snow, fog, heavyrain, downpour, rainstorm, heavyrains, snowfall, rainy, mist | rain | glaw (*rain*), eira (*snow*), cenllysg (*hail*), wlith (*dew*), cawodydd (*showers*), rhew (*frost*), taranau (*thunder*), barrug (*frost*), genllysg (*hail*), dafnau (*drops*) |
| hapus (*happy*) | happy, pleased, glad, grateful, delighted, thankful, anxious, eager, fortunate, confident | happy | hapus (*happy*), bodlon (*satisfied*), ffeind (*kind*), llawen (*joyful*), rhyfedd (*strange*), trist (*sad*), llon (*cheerful*), cysurus (*comfortable*), hoenus (*cheerful*), nerfus (*nervous*) |
| meddalwedd (*software*) | software, application, computer, system, tool, ibm, hardware, technology, database, device | software | meddalwedd (*software*), feddalwedd (*software*), caledwedd (*hardware*), dyfeisiau (*devices*), amgryptio (*encryption*), dyfeisiadau (*inventions*), cymwysiadau (*applications*), algorithm (*algorithm*), dyfais (*device*), ategyn (*plugin*) |
| ffrangeg (*French language*) | Arabic, Hebrew, Hindi, Arabiclanguage, language, urdu, sanskrit, haitiancreole, English | French | ffrengig (*french*), sbaenaidd (*Spanish*), almaeneg (*German language*), archentaidd (*Argentinian*), gwyddelig (*Irish*), twrcaidd (*Turkish*), llydewig (*Breton*), almaenaidd (*German*), danaidd (*Danish*), imperialaidd (*imperial*) |
| croissant (*croissant*) | frenchfries, yogurt, applesauce, currysauce, mulledwine, mozzarellacheese, noodlesoup, buñuelo, chilisauce, misosoup | croissant | bisgedi (*biscuits*), twmplenni (*dumplings*), byns (*buns*), teisennau (*cakes*), bacwn (*bacon*), caramel (*caramel*), melysfwyd (*confectionery*), cwstard (*custard*), marmaled (*marmalade*) |
| gwario (*spend money*) | expend, invest, reinvest, pay, allot, allocate, disburse, economize, retrench, accrue | spend | treulio (*spend time*), dreulio (*spend time*), gwario (*spend money*), aros (*wait*), threulio (*spend time*), gwastraffu (*wasting*), dychwelyd (*returning*), nychu (*languishing*), hala (*spend money*), byw (*live*) |
| hiraeth (*longing*) | longing, sadness, yearning, sorrow, anguish, loneliness, grief, feeling, ennui, heartache | longing | hiraeth (*longing*), galar (*grief*), anwyldeb (*dearness*), tristwch (*sadness*), nwyd (*passion*), gorfoledd (*exultation*), tosturi (*compassion*), tynerwch (*tenderness*), nwyf (*vivacity*), hyfrydwch (*loveliness*) |
| caerdydd (*Cardiff*) | Docklands, Southbank, Brisbane, Frankston, downtown, Thessaloniki, Coquitlam, Melbourne, Bayside, Glasgow | Cardiff | Abertawe (*Awansea*), nantporth (*Nantporth*), aberystwyth (*Aberystwyth*), llanelli (*Llanelli*), glynebwy (*Ebbw Vale*), caerdydd (*Cardiff*), porthcawl (*Porthcawl*), llandudno (*Llandudno*), awyr (*sky*), wrecsam (*Wrexham*) |
| Tonypandy (*Tonypandy*) | Edgewareroad, Blakelaw, Upperdicker, Ainleytop, Bilsthorpe, Romanby, Killay, Llanwonno, Penllergaer, Greenrigg | Tonypandy | aberdâr (*Aberdare*), Senghennydd (*Senghennydd*), aberpennar (*Mountain Ash*), brynaman (*Brynamman*), aberdar (*Aberdare*), coedpoeth (*Coedpoeth*), llanidloes (*Llanidloes*), brynbuga (*Usk*), penycae (*Penycae*), tymbl (*Tumble*) |

### 5.3. Extrinsic Evaluation

The extrinsic evaluation assesses the performance of a language model in the context of a predefined task. In this study, this task was chosen to be that of sentiment analysis (SA), as it has been shown that cross-lingual systems can achieve high accuracy even in zero or few-shot settings [4]. Specifically, given the shortage of annotated Welsh corpora that can be used to train a Welsh SA model, we wanted to investigate to what extent cross-lingual embeddings can improve the performance of such a model by re-using a readily available annotated English data set.

To implement SA, we re-purposed an existing sentence classifier [51] based on a convolutional neural network for text classification [52], which has been extended by a bidirectional long–short-term memory (Bi-LSTM) [53] layer. This classification model is well equipped to capture both short- and long-range dependencies and extract general features of online reviews that would be useful for SA. The most important hyperparameters of the base model include 100 convolutional filters, a kernel of size 4 and strides of size 1, with a ReLu activation function. Further, the Bi-LSTM layer consisted of two 100-unit (forward and backward) LSTM layers. The model was trained using categorical cross-entropy with an Adam optimizer. In this model, each training instance is represented as a matrix, where each word is represented by the corresponding embedding. Such representation is suitable for cross-lingual training, as cross-lingual synonyms are expected to be represented by similar vectors in the joint vector space. Therefore, any abstractions learned by the model are also expected to be similar in the two languages.

All SA experiments were performed using a set of 50 K IMDB reviews, which represent a community standard for evaluating SA [54]. This data set is divided into two subsets of 25 K reviews, each to be used for training and testing, respectively. The original reviews were automatically translated from English to Welsh using Google Translate, a neural machine translation system [55] that proved mature enough to produce reliable data for training SA in languages other than English [56]. We used the best-performing bilingual English–Welsh embeddings as per their performance in the dictionary induction task (Section 5.1).

To perform cross-lingual training, we started by training an SA model using English data only and evaluated the results using Welsh data. We call this zero-short learning as no labeled data in Welsh were used at all. We then gradually added Welsh translations using increments of $n$ reviews, where $n$ = 100, 500, 1000, 2500, 5000, 7500, 10,000, 12,500, and 150,00. Given a fixed size $n$, a random subset was selected five times to check whether the evaluation results were reproducible. All experiments were evaluated against the Welsh test data.

Figure 1 shows the evaluation results in terms of accuracy (y axis) against exposure to labeled data in Welsh (x axis refers to the total number of reviews of Welsh that were combined with a total of 25 K reviews in English). The zero-shot model achieves an accuracy of 65%. The accuracy increased substantially by adding as little as a thousand reviews automatically translated to Welsh. Naturally, with increased exposure to Welsh during the training; the accuracy increased as well. Already at 5000 Welsh reviews, the average accuracy surged beyond 75%. In addition, the model stabilized as less variance was observed across the experiments using different subsets of a fixed size. The highest accuracy achieved fell just short of 80%. Further performance gains are expected to be obtained by tuning the hyperparameters or the neural network architecture itself to optimize its performance with Welsh. However, this is well beyond the scope of the current study. Nonetheless, our experiments confirmed that cross-lingual embeddings make zero-shot English-to-Welsh SA possible with few-shot settings contributing to considerable performance improvements. These results provide the evidence that existing NLP tools based on word embeddings can indeed be re-used to support NLP in Welsh.

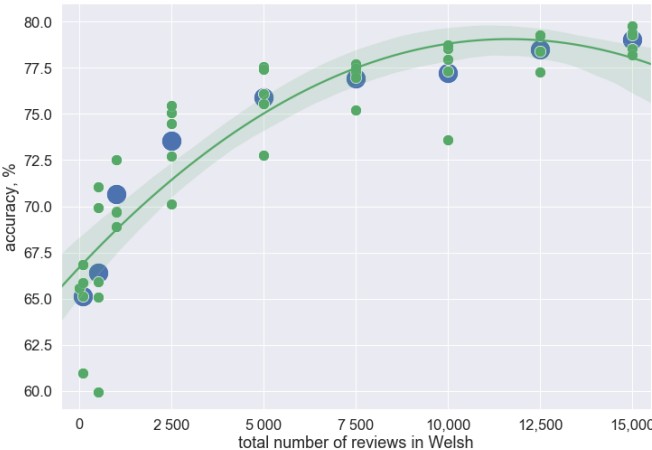

**Figure 1.** Accuracy results for the cross-lingual sentiment analysis experiment.

## 6. Conclusions

We have described the process of training bilingual English–Welsh embeddings. We start by discussing the corpora we used to train monolingual embeddings in both languages using both *word2vec* and *fastText*, and continue by explaining the curation of the supervision signal (the training bilingual dictionary), as well as the linear transformation method we use for mapping both monolingual spaces into a shared bilingual space.

We have evaluated this shared space both intrinsically and extrinsically. The intrinsic evaluation was based on *dictionary induction*, which was used to measure the alignment of two monolingual spaces directly by translating between the two languages and measuring the distance within a monolingual space. The best alignment was achieved by training the monolingual spaces using fastText and aligning them using the the CSLS metric. The true value of aligning two vector spaces lies in the ability to facilitate NLP applications in minoritized languages by taking advantage of readily available resources in a language such as English. To evaluate the cross-lingual embeddings extrinsically, we measured the effects of supplementing Welsh-language data with data in English on the accuracy of sentiment analysis in Welsh. We were able to use an existing neural network architecture based on CNNs and LSTMs originally developed for sentiment analysis in English. By training this neural network on cross-lingual embeddings and data from both languages, we managed to obtain highly competitive results in Welsh without having to modify the original method in any way. In particular, we demonstrated that a relatively small data set of 2 K documents in the target language seems to suffice. This opens exciting avenues for future work, where cross-lingual embeddings can be combined with neural architectures and data augmentation techniques to develop Welsh language technology at a negligible cost.

The Welsh language can be categorized, within the language resource landscape, as being a low-resource language, i.e., the availability of (raw and annotated) corpora, glossaries, thesauri, encyclopedias, etc. is limited when compared to other languages such as English, Chinese, Spanish, or Indo-Aryan languages. This study allows one to automatically compare the meaning of words not only within the Welsh language but also across the two languages, thus facilitating applications such as the creation of bilingual language resources, as well as the development of NLP systems for Welsh with limited Welsh training data, as we successfully demonstrated with sentiment analysis. Cross-lingual embedding we generated therefore unlocks access to a plethora of open-source NLP solutions developed originally for English. This in turn opens a possibility of supporting a wide range of applications, such as computer–assisted translation, cross-lingual information retrieval, and conversational artificial intelligence. These applications encourage the use of Welsh in activities of daily life, which contributes to maintaining and improving Welsh language skills.

**Author Contributions:** Conceptualization, D.K.; methodology, L.E.-A., G.P., P.C., M.F., and I.S.; software, L.E.-A., G.P., M.F., and I.S.; validation, I.S. and D.K.; formal analysis, L.E.-A., G.P., and I.S.; investigation, L.E.-A., G.P., and I.S.; resources, L.E.-A., G.P., P.C., I.S., and D.K.; data curation, L.E.-A., G.P., P.C., I.S., and D.K.; writing—L.E.-A.; writing—review and editing, L.E.-A., G.P., P.C., I.S., and D.K.; visualization, M.F.; supervision, L.E.-A., I.S., and D.K.; project administration, P.C., I.S., and D.K.; funding acquisition, I.S. and D.K. All authors have read and agreed to the published version of the manuscript.

**Funding:** This research was funded by the Welsh Government, under the Grant "Learning English-Welsh bilingual embeddings and applications in text categorisation".

**Institutional Review Board Statement:** Not applicable.

**Data Availability Statement:** Data and software to reproduce our results are available at: https://github.com/cardiffnlp/en-cy-bilingual-embeddings, accessed on 15 July 2021.

**Acknowledgments:** The research on which this article is based was funded by the Welsh Government as part of the "Learning English–Welsh bilingual embeddings and applications in text categorisation" project.

**Conflicts of Interest:** The funders had no role in the design of the study; in the collection, analyses, or interpretation of data; in the writing of the manuscript, or in the decision to publish the results.

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
