# Peer review of "English–Welsh Cross-Lingual Embeddings"

_applsci, doi:10.3390/app11146541_

Round 1
Reviewer 1 Report
The paper presents an interesting study on cross-lingual embeddings for the English-Welsh language pair. The paper is pretty well written and structured and the study follows a sound methodological approach that appears to be rigorously applied.
Some minor amendments, however, should be done to improve the quality of the presentation:
- The abstract lacks some brief reference to the methodology applied. I would suggest including 1-2 sentences in this respect.
- The conclusion section seems rather brief. I would suggest adding some extra information about the implications of this study for the research field.
Once these minor issues are amended, the paper will be suitable for publication in this journal.
Author Response
A1. The abstract lacks some brief reference to the methodology applied. I would suggest including 1-2 sentences in this respect.
|
We re-wrote the abstract as follows:
Cross-lingual embeddings are vector space representations where word translations tend to be co-located in this space. These representations enable transfer learning across languages, thus bridging the gap between data-rich languages such as English and others. In this paper we present and evaluate a suite of cross-lingual embeddings for the English-Welsh language pair. To train the bilingual embeddings, a Welsh corpus of approximately 145M words was combined with English Wikipedia corpus. We used a bilingual dictionary to frame the problem of learning bilingual mappings as a supervised machine learning task, where a word vector space is first learned independently on a monolingual corpus, after which a linear alignment strategy is applied to map the monolingual embeddings to a common bilingual vector space. Two approaches were used to learn monolingual embeddings including word2vec and fastText. Three cross-language alignment strategies were explored including cosine similarity, inverted softmax and cross-domain similarity local scaling (CSLS). We evaluated different combinations of these approaches using two tasks, bilingual dictionary induction and cross-lingual sentiment analysis. The best results were achieved using monolingual fastText embeddings and the CSLS metric. We also demonstrated that by including a few automatically translated training documents, the performance of a cross-lingual text classifier for Welsh can increase by approximately 20 percent points. |
A2. The conclusion section seems rather brief. I would suggest adding some extra information about the implications of this study for the research field.
|
We expanded the conclusions as follows:
We have described the process of training bilingual English-Welsh embeddings. We start by discussing the corpora we used to train monolingual embeddings in both languages using both word2vec and fastText, and continue by explaining the curation of the supervision signal (the training bilingual dictionary), as well as the linear transformation method we use for mapping both monolingual spaces into a shared bilingual space.
We have evaluated this shared space both intrinsically and extrinsically. The intrinsic evaluation was based on dictionary induction, which was to measure the alignment of two monolingual spaces directly by translating between the two languages and measuring the distance within a monolingual space. The best alignment was achieved by training the monolingual spaces using fastText and aligning them using the the CSLS metric. The true value of aligning two vector spaces lies in the ability to facilitate NLP applications in minoritised languages by taking advantage of readily available resources in a language such as English. To evaluate the cross-lingual embeddings extrinsically, we measured the effects of supplementing Welsh-language data with data in English on the accuracy of sentiment analysis in Welsh. We were able to use an existing neural network architecture based on CNNs and LSTMs originally developed for sentiment analysis in English. By training this neural network on cross-lingual embeddings and data from both languages, we managed to obtain highly competitive results in Welsh without having to modify the original method in any way. In particular, we demonstrated that a relatively small data set of 2K documents in the target language seems to suffice. This opens exciting avenues for future work, where cross-lingual embeddings can be combined with neural architectures and data augmentation techniques to develop Welsh language technology at a negligible cost.
The Welsh language can be categorized, within the language resource landscape, as being a low-resource language, i.e. the availability of (raw and annotated) corpora, glossaries, thesauri, encyclopedia, etc. is limited when compared to other languages such as English, Chinese, Spanish or Indo Aryan languages. This study allows one to automatically compare the meaning of words not only within the Welsh language but also across the two languages, thus facilitating applications such as creation of bilingual language resources, as well as development of NLP systems for Welsh with limited Welsh training data as we successfully demonstrated with sentiment analysis. Cross-lingual embedding we generated therefore unlock access to a plethora of open-source NLP solution developed originally for English. This in turn opens a possibility of supporting a wide range of applications such as computer–assisted translation, cross-lingual information retrieval and conversational artificial intelligence. These applications encourage the use of Welsh in activities of daily life, which contributes to maintaining and improving Welsh language skills. |
Reviewer 2 Report
This manuscript is to present cross-lingual embedding focused on between resource-rich and -less languages, eg. English and Welsh as shown in this manuscript. This manuscript is well-designed and described relatively, but needed to improve for readers as follows:
(1) Bilingual lexicon induction is definitely used for rare words like hapax legomena rather than frequent words, thus the qualitative and/or quantitative analysis of rare words should be required in more detail.
(2) Accuracy is good evaluation metric for bilingual lexicon induction, but mean reciprocal rank (MRR), precision, and recall are also widely used. Please see the following references:
A. Irvine and V. Callison-Burch, A Comprehensive analysis of bilingual lexicon induction, Computational Linguistics 43(2):273-310, 2017.
J.-H. Kim, H.-S. Kwon, and H.-W. Seo, Evaluating a pivot-based approach for bilingual lexicon extraction, Computational Intelligence and Neuroscience 2015:1-13, 2015.
(3) Literature reviews like MUSE(https://github.com/facebookresearch/MUSE ), and so on are needed in more detail as a background.
(4) On line 273, there is a typo (describe din -> described in).
(5) In Table 2, there are typos(fasttext -> fastText; csls -> CSLS)
(6) On line 306, conjouring (???)
(7) In Figure 1, what does the x axis represent correctly? the number of reviews? or the number of tokens of English or Welsh? It is little bits confusing.
Author Response
B1. Bilingual lexicon induction is definitely used for rare words like hapax legomena rather than frequent words, thus the qualitative and/or quantitative analysis of rare words should be required in more detail.
|
This is an interesting point. However, the major impact on performance of NLP application using cross-lingual embeddings would depend on frequently occurring words (e.g. text classification), so in the respect, the analysis of rare words may be out of scope of this particular study, but is certainly worthy of reporting in a separate publication. |
B2. Accuracy is good evaluation metric for bilingual lexicon induction, but mean reciprocal rank (MRR), precision, and recall are also widely used. Please see the following references:
Irvine and V. Callison-Burch, A Comprehensive analysis of bilingual lexicon induction, Computational Linguistics 43(2):273-310, 2017.
J.-H. Kim, H.-S. Kwon, and H.-W. Seo, Evaluating a pivot-based approach for bilingual lexicon extraction, Computational Intelligence and Neuroscience 2015:1-13, 2015.
|
Thanks for the suggestion. We will certainly consider these measures in future papers focusing primarily on bilingual lexicon induction. At present, accuracy served the goal of comparing different cross-lingual embedding approaches to one another well, so there is no strong justification to explore different measures on this particular occasion. |
B3. Literature reviews like MUSE (https://github.com/facebookresearch/MUSE), and so on are needed in more detail as a background.
|
We have expanded the background as follows:
Earlier attempts to train cross-lingual word embeddings required access to parallel (or, at least, comparable) corpora \cite{klementiev2012inducing, zou2013bilingual, kneser1995improved, lauly2014autoencoder, kovcisky2014learning, coulmance2016trans, wang2016novel}. Finding such corpus especially for a minoritised language can prove challenging. Therefore, the research in this space gravitated towards using bilingual dictionaries instead to align the words in respective languages \cite{mikolov2013exploiting, faruqui2014improving}. It was later shown that such cross-lingual supervision is not necessary to align word embedding \cite{conneau2018word}. Instead, adversarial training can be used to initialise a linear mapping between two vector spaces and produce a synthetic parallel dictionary. The success of this approach was based on the use of two metrices, one for unsupervised validation and the other one for similarity measure. Such combination reduces the hubness problem while improving the translation accuracy.
Hubness is a phenomenon that occurs in high-dimensional spaces, where some objects tend to concentrate around a centroid while others have few nearest neighbours \cite{radovanovic2010hubs}. Specifically, hubness associated with cross-lingual embeddings was explored in \cite{dinu2015improving} who proposed incorporating a nearest neighbour reciprocity as a way of managing hubness. Different measures were used to down-weigh similarities associated with hub words including cross-domain similarity local scaling (CSLS) \cite{conneau2018word} and inverted softmax \cite{smith2017offline}. In addition, adding an orthogonality constraint, which conveniently has a closed-form solution, can improve performance the performance further \cite{xing2015normalized}.
Alternatively, to align monolingual embedding spaces with no supervision, Zhang et al. \cite{zhangb} used adversarial training to exploit sudden drops in accuracy for model selection followed by minimising the earth-mover distance \cite{zhanga}. Conversely, Conneau et al. \cite{conneau2018word} do not base model selection on its performance, which allows for hyper-parameters to be tuned specifically for a given language pair as these do tend to vary significantly across languages.
Similar approaches were used to induce bilingual dictionaries from data \cite{artetxe-labaka-agirre:2017:Long, adams2017cross, artetxe-etal-2018-robust, doval2018improving} yields state-of-the-art performance in many language pairs, although the experimental setup followed in the literature has also been closely scrutinized, and there exist studies that argue for experiments that account for different genres in source and target corpora, studying (dis)similarities between languages, etc. \cite{sogaard2018limitations, doval2020robustness}.
Although the advent of language models in the current NLP landscape (BERT, GPT or RoBERTa) \cite{devlin2018bert, radford2019language, liu2019roberta} has transformed the field, it is also true that for languages where even availability of raw data is small, having access to pre-trained static word embeddings can make the difference between developing a language technology at all or not. Recent work has, for example, focused on dialectal Arabic, by combining BERT-based encodings with Arabic word embeddings for underrepresented domains and dialects \cite{alghanmi2020combining}. |
B4. On line 273, there is a typo (describe din -> described in).
|
Corrected. |
B5. In Table 2, there are typos(fasttext -> fastText; csls -> CSLS)
|
All references to fastText have now been corrected. CSLS is now upercased. |
B6. On line 306, conjouring (???)
|
We replaced "conjouring" by "referring to". |
B7. In Figure 1, what does the x axis represent correctly? the number of reviews? or the number of tokens of English or Welsh? It is little bits confusing.
|
We added the following explanation:
"x axis refers to the total number of reviews of Welsh that were combined with a total of 25K reviews in English"
We also updated the x axis to refer to Welsh reviews only as these were previously conflated with English reviews, which was somewhat confusing. We renamed the x axis to "total number of reviews in Welsh". We explicitly refer to reviews, so together with the text description, it should now be clear that the number refers to documents and not to tokens. |